# Impact of Endoluminal Radiofrequency Ablation on Immunity in Pancreatic Cancer and Cholangiocarcinoma

**DOI:** 10.3390/biomedicines10061331

**Published:** 2022-06-06

**Authors:** Jana Jarosova, Peter Macinga, Lenka Krupickova, Martina Fialova, Alzbeta Hujova, Jan Mares, Ondrej Urban, Jan Hajer, Julius Spicak, Ilja Striz, Tomas Hucl

**Affiliations:** 1Department of Gastroenterology and Hepatology, Institute for Clinical and Experimental Medicine, Videnska 1958, 140 21 Prague, Czech Republic; jana.jarosova@ikem.cz (J.J.); peter.macinga@ikem.cz (P.M.); alzbeta.hujova@ikem.cz (A.H.); julius.spicak@ikem.cz (J.S.); 2Department of Clinical and Transplant Immunology, Institute for Clinical and Experimental Medicine, Videnska 1958, 140 21 Prague, Czech Republic; lenka.krupickova@ikem.cz (L.K.); martina.fialova@ikem.cz (M.F.); ilja.striz@ikem.cz (I.S.); 3Department of Data Analysis, Statistics and Artificial Intelligence, Institute for Clinical and Experimental Medicine, Videnska 1958, 140 21 Prague, Czech Republic; jan.mares@ikem.cz; 4Department of Internal Medicine II—Gastroenterology and Hepatology, University Hospital Olomouc and Faculty of Medicine and Dentistry, Palacky University Olomouc, I.P. Pavlova 185/6, 779 00 Olomouc, Czech Republic; ondrej.urban@fnol.cz; 5Department of Internal Medicine, 3rd Faculty of Medicine, Charles University, University Hospital Kralovske Vinohrady, Srobarova 1150, 100 34 Prague, Czech Republic; jan.hajer@fnkv.cz

**Keywords:** pancreatic ductal adenocarcinoma, cholangiocarcinoma, radiofrequency ablation, antitumor immunity

## Abstract

Radiofrequency ablation (RFA) is a mini-invasive loco-regional ablation technique that is increasingly being used as a palliative treatment for pancreatic cancer and cholangiocarcinoma. Ablation-triggered immune system stimulation has been proposed as a mechanism behind the systemic effects of RFA. The aim of our study was to investigate the immune response to endoluminal biliary RFA. Peripheral blood samples were collected from patients with pancreatic cancer and cholangiocarcinoma randomised to receive endoluminal biliary radiofrequency ablation + stent (19 patients) or stent only (21 patients). We observed an early increase in IL-6 levels and a delayed increase in CXCL1, CXCL5, and CXCL11 levels as well as an increase in CD8+ and NK cells. However, these changes were not specific to RFA treatment. Explicitly in response to RFA, we observed a delayed increase in serum CXCL1 levels and an early decrease in the number of anti-inflammatory CD206+ blood monocytes. Our study provides the first evidence of endoluminal biliary RFA-based regulation of the systemic immune response in patients with pancreatic cancer and cholangiocarcinoma. These changes were characterised by a general inflammatory response. RFA-specific activation of the adaptive immune system was not confirmed.

## 1. Introduction

Pancreatic cancer (PC) and perihilar cholangiocellular carcinoma (CCC) are malignancies with increasing incidence, representing the main causes of malignant biliary obstruction. Late diagnosis preventing surgical resection as well as chemo- and radio-resistance are factors that result in an overall five-year survival not exceeding 10%. Therefore, there is a pressing need for therapeutic alternatives [1].

Radiofrequency ablation (RFA) is a local ablative technique routinely applied in the treatment of numerous neoplasia including Barrett’s oesophagus-related dysplasia and hepatocellular carcinoma [2,3]. Percutaneous or intraoperative needle-based RFA has been used in pancreatic cancer [4,5]. The developments in catheter-based RFA means pancreatic cancer and cholangiocarcinoma can now be treated via endoluminal application in the bile duct, possibly also improving biliary stent patency. Three recent randomised controlled studies, numerous retrospective studies, and two meta-analyses have demonstrated the significant effect of endoluminal RFA on prolonging survival as well as stent patency [3,6,7,8,9,10].

The high-frequency current used in RFA induces hyperthermic cell injury, resulting in coagulative necrosis and cell death. Necrosis produces a loss of plasma membrane integrity and the release of necrotic-cell intracellular antigens as well as damage-associated molecular patterns (DAMP), including heat shock proteins (HSPs), parts of intracellular organelles, and RNA/DNA particles. These debris components contain tumour antigens that can be recognised and targeted by the host immune system, leading to its activation [11]. The ablated area becomes densely populated by neutrophils, macrophages, dendritic cells (DC), natural killer cells (NK), and B and T lymphocytes, which increase cytokine production [12,13]. According to several animal and human studies, the immune response is not limited to the local site of ablation, and a systemic reaction may also be present [14].

Apart from the local immune response, there is a growing consensus that RFA also exerts a systemic effect. Multiple cytokines, including IL-1β, IL-6, IL-8, and TNF-α, remain elevated hours to days after ablation [15]. Heat shock protein 70 (HSP70) is understood to play a key role in mediating the immunostimulatory effect of RFA by serving as an antigen chaperone to antigen-presenting cells or by activating DCs. In a study by Haen et al., significant elevation of serum HSP70 one day after RFA was associated with favourable clinical outcomes [16]. Furthermore, a systemic antitumour T-cell immune response, characterised by increased levels of CD8+ and CD4+ T cells, has been repeatedly observed [17,18,19].

Only one study to date has documented the immune effects of RFA in human pancreatic cancer. Giardino et al. analysed peripheral blood samples in 10 patients with locally advanced pancreatic cancer before and after intraoperative RFA. They found a transient increase in levels of IL-6 and a permanent increase in CD8+ and CD4+ T lymphocytes and DCs, indicating true immunomodulation rather than a normal inflammatory response [14].

In addition, there is indirect clinical evidence for the systemic effect of radiofrequency ablation. The regression of metastases following RFA of a primary renal cell carcinoma has been documented [20] as well as the delayed growth of distant untargeted hepatocellular carcinoma liver nodules via a mechanism dependent on systemic CD8+ T-cell-mediated antitumour immunity [21].

Our aim was to investigate the impact of endoluminal biliary radiofrequency ablation on the systemic immune response in pancreatic cancer and cholangiocarcinoma using a well-controlled human model.

## 2. Materials and Methods

### 2.1. Patients

This study drew on a large multicentre randomised controlled trial (RFA for malignant biliary obstruction, NCT03166436) investigating the role of endoluminal biliary RFA on survival and stent patency in supplementation with stenting in patients with pancreatic cancer and cholangiocarcinoma. For the purpose of our study, we analysed the immune system response in a proportion (25%) of consecutive patients enrolled in the main study at the Institute for Clinical and Experimental Medicine, Prague.

Inclusion criteria were age over 18, histologically confirmed pancreatic cancer or cholangiocarcinoma, laboratory signs of biliary obstruction, nonresectable disease, and life expectancy greater than 3 months. Exclusion criteria were instability for endoscopic procedure, pregnancy, uncontrolled coagulopathy, concomitant biliary oncological endoluminal therapy, and cardiac pacemaker. None of the patients enrolled in our immune response study received chemotherapy or other concomitant oncological treatment, and they were not on corticosteroids or immunosuppressive medication.

The study was approved by the local ethics committee and performed according to the Declaration of Helsinki, incorporating changes accepted in Seoul, South Korea, during the 59th WMA General Assembly.

### 2.2. Radiofrequency Ablation

Radiofrequency ablation was delivered endoluminally during endoscopic retrograde cholangiography (ERC). All patients underwent either general anaesthesia or conscious sedation with midazolam and fentanyl. Following bile duct cannulation and opacification, a Habib bipolar 8Fr catheter (Boston Scientific, Marlborough, MA, USA) was introduced over a wire into the stricture. Ablation was administered via an ERBE power generator using effect 8 at 10 W for 120 s. Either one or two sessions (depending on the anatomy) of ablation were performed to cover the entire area of the stricture. Following ablation, either a single stent or multiple metal (whenever anatomy allowed) or plastic stents were placed. In patients randomised to the control group, stent placement was performed without ablation. All patients were administered antibiotics before and on the first day after the procedure (ciprofloxacin 200 mg intravenously).

### 2.3. Blood Collection

Peripheral blood from patients was collected in EDTA tubes at four time points: before the procedure and then 1, 14, and 30 days after endoscopy. From whole blood samples, the expression of membrane antigens on blood cells was measured by flow cytometry. Additional blood samples were centrifuged, with plasma separated into new tubes. Plasma was frozen at −80 °C and subsequently used to detect soluble markers.

### 2.4. Flow Cytometry

Peripheral blood cells were labelled with fluorochrome-conjugated monoclonal antibodies and resuspended for 20 min at room temperature (RT) in the dark. Red blood cells were lysed by Optilyse C (Beckman Coulter, Brea, CA, USA) for 10 min. The reaction was stopped by CellWash buffer (BD, Franklin Lakes, NJ, USA). The monoclonal antibodies used in the study are summarised in Table 1.

Samples were measured on the Navios EX flow cytometer (Beckman Coulter, Brea, CA, USA) and analysed using Navios and Kaluza software (Beckman Coulter, Brea, CA, USA). The percentage of positive cells as well as median/mean fluorescence intensity (MFI) were evaluated for each individual marker. Subpopulations of blood monocytes were discriminated as classical (CD14+CD16−), intermediate (CD14+CD16+), and nonclassical (CD14lowCD16+). For the gating strategy, see Figure 1.

### 2.5. Measurement of Soluble Markers in Plasma Using ELISA and Luminex Technology

Soluble markers were detected using ELISA and Luminex multiplex technology. Concentrations of HSP70 were detected using the DuoSet ELISA kit (R&D, Minneapolis, MN, USA) and concentrations of HMGB-1 using the appropriate ELISA kit (NOVUS Biologicals, Denver, CO, USA). For the detection of other cytokine and chemokine (a panel of 18 markers is summarised in Table 2) concentrations, a magnetic Luminex multi-analyte kit was used (R&D, Minneapolis, MN, USA).

### 2.6. Luminex Assay Procedure

All reagents from both the samples and the preconfigured kit were prepared. First, 50 µL of either standards or samples were added per well together with 50 µL of microparticle cocktail. The microplate was then incubated for 2 h at room temperature on a horizontal orbital shaker set at 800 rpm. Each well was filled with 100 µL of Wash Buffer, with the wash procedure performed three times in total. Next, 50 µL of diluted Biotin-Antibody Cocktail was added to each well and incubated for 1 h at room temperature on a horizontal orbital microplate shaker set at 800 rpm. Again, three wash cycles were performed. Fifty microlitres of diluted Streptavidin–PE was added to each well and incubated for 30 min at room temperature on a horizontal orbital microplate shaker set at 800 rpm. After washing, the microparticles were then resuspended by adding 100 µL of Wash Buffer to each well. Finally, the plate was incubated for 2 min at room temperature on a horizontal orbital microplate shaker set at 800 rpm. The microplate was read using a LABScan 3D instrument (Luminex Corporation, One Lamba, Los Angeles, CA, USA).

### 2.7. ELISA R&D DuoSet Assay Procedure

The diluted capture antibody was coated in a 96-well microplate (100 µL per well) and incubated overnight. Each well was washed three times using 400 µL of Wash Buffer. After washing, the plate was blocked using 300 µL of Reagent Diluent for 1 h. The wash procedure was then repeated. One hundred microlitres of standards and samples were added and incubated for two hours. Again, the washing procedure was repeated. 100 µL of diluted detection antibody was added and incubated for two hours, repeating the washing procedure. One hundred microlitres of streptavidin–HRP was added and incubated for 20 min in the dark, once again repeating the washing procedure. One hundred microlitres of Substrate solution was added and incubated for 20 min in the dark. The reaction was stopped with the addition of 50 µL of Stop solution. The optical density was measured using the ELISA Sunrise spectrophotometer (Tecan, Männedorf, Switzerland) set to 450 nm, with the results evaluated using KIM software.

### 2.8. ELISA NOVUS Biologicals Assay Pro

First, 100 µL of standards and samples were added to a 96-well microplate and incubated for 90 min at 37 °C. The liquid was then removed followed by the addition of 100 µL of biotinylated detection antibody incubated for 1 h at 37 °C. Each well was washed four times with 350 µL of Wash Buffer followed by the addition of 100 µL of HRP conjugate incubated for 30 min at 37 °C. The wash procedure was repeated, and 90 µL of Substrate reagent was added and incubated for 15 min at 37 °C in the dark. The reaction was stopped by adding 50 µL of Stop solution. The optical density was measured using the ELISA Sunrise spectrophotometer (Tecan, Männedorf, Switzerland) set to 450 nm, with the results evaluated on KIM software.

### 2.9. Public Database Analysis

For the analysis of tumour cytokine expression and its correlation with prognosis, the cBioPortal tool [22] containing data sets of pancreatic cancer and cholangiocarcinoma from the Cancer Genome Atlas was used. Further, the association of tumour tissue cytokine expression with immune cell infiltration was analysed using the TIMER analysis tool [23].

### 2.10. Statistical Analysis

In the uncontrolled analysis, we used the nonparametric Wilcoxon signed-rank test to test for a difference between the baseline and each of the following visits. In the controlled analysis, the nonparametric Brunner–Munzel test was used to determine if the absolute change from baseline differed between the experimental group and the control group. In both cases, the two-sided alternative was used. The false discovery rate (FDR) approach by Benjamini and Hochberg was used to correct for multiple testing in sets of tests (i.e., visits, tumour types, and marker groups). A threshold of 0.05 for resulting q-values was used to indicate statistical significance. Therefore, only 5% of the results reported as significant are expected to be false positives. The software packages “lawstat” (v.3.4), “fuzzySim” (v.4.0), and “ggplot2” (v.3.3.5) for R (v. 4.1.3) were used for the statistical analysis.

## 3. Results

In total, 40 consecutive patients with malignant biliary obstruction were included in the study. Of these, 19 patients had pancreatic cancer and 21 patients had cholangiocarcinoma. Patients were randomised to either endoluminal radiofrequency ablation + stenting (RFA group, *n* = 19) or stenting alone (controls, *n* = 21). The group comprised 25 men and 15 women with a mean age of 68.5 years. None of the patients developed peri-procedural or post-procedural (within 30 days of the procedure) complications. Detailed characteristics of the patients are given in Table 3.

Immune response was evaluated based on the level of circulating cytokines and the number of immune cells in peripheral blood. The results obtained were evaluated in two ways. First, we assessed only those changes induced in patients receiving RFA (i.e., uncontrolled model). Second, we compared these changes to those occurring in the control group (i.e., controlled model).

### 3.1. Role of Immune System in Pancreatic Cancer and Cholangiocarcinoma

In order to support the importance of the immune system in pancreatic cancer and cholangiocarcinoma and the relevance of the selected cytokines, we analysed their expression in tumour tissue using the cBioPortal tool. We found an association of increased expression of CXCL9, CXCL10, and CXCL10 with decreased survival in pancreatic cancer and CXCL5 in cholangiocarcinoma (Appendix A). Furthermore, we investigated whether expression of the selected cytokines was correlated with immune cell infiltration using the TIMER tool, confirming the role of some of the cytokines in attracting immune cells into the site of the tumour in pancreatic cancer and cholangiocarcinoma (Appendix A).

### 3.2. Impact of RFA on Circulating Cytokines and Chemokines

#### 3.2.1. Uncontrolled Model

We first investigated the impact of RFA on the levels of soluble cytokines and chemokines. There was a clear significant increase in IL-6 on day 1 in all patients. These values returned to baseline on day 14, remaining unchanged until day 30. In all patients, we also found a significant increase in CXCL11 levels on day 14, which continued until day 30. In contrast, an observed increase in CXCL1 on day 14 disappeared by day 30. There was a marked and significant rise in CXCL5 levels on day 14 in all patients, which continued until day 30 (Figure 2).

#### 3.2.2. Controlled Model

We also wanted to determine which of these changes would remain significant. To that end, we compared the levels of cytokines and chemokines to levels measured in patients not treated with RFA. The only difference that remained significant after controlling against the RFA-untreated group was the rise in CXCL1 levels. Chemokine levels were significantly elevated on day 14 only in patients treated with RFA. The other changes observed in the RFA-treated group, namely, the transient rise in IL-6 on day 1 and the permanent rise in CXCL11 and CXCL5 from day 14 onward, proved nonsignificant when compared to the control group (Figure 3).

### 3.3. Impact of RFA on Peripheral Circulating Immune Cells

#### 3.3.1. Uncontrolled Model

In patients treated with RFA, there was an increase in lymphocyte numbers on day 14, which continued (despite reducing) until day 30. Specifically, we observed an increase in total T cells (CD3+ cells) and a decrease in total B cells (CD19+ cells). There was a statistically significant rise in the absolute number of CD8+ cells on day 14, which continued until day 30. The number of NK cells also rose on day 14, remaining elevated until the end of the study (Figure 4).

#### 3.3.2. Controlled Model

When controlling cell population changes against RFA-untreated patients, we found that similar changes occurred in the control group, rendering them non-significant. A mild but significant decrease in the number of CD 206+ cells observed on day 1 later normalised during the course of treatment. The number of CD47+ cells increased in the control group after the ERCP procedure until the end of the observation period compared to a decrease in RFA-treated cells. The most significant difference was observed on day 14. All other changes observed in treated patients following the RFA procedure proved insignificant compared to untreated patients (Figure 5).

## 4. Discussion

Radiofrequency ablation is a thermal ablative method for palliative cancer treatment, producing local coagulative necrosis. Recent studies point to the role of RFA in prolonging the survival of patients with malignant biliary obstruction [6,7,9,10], and numerous other works involving animal and human models highlight its effect on antitumour immunity [12,13,21]. Our study provides novel evidence of the impact of endoluminal RFA in patients with pancreatic cancer and cholangiocarcinoma on the systemic immune system. In our well-controlled human model, the changes observed in our patients were compatible with a general inflammatory response. We also recorded prolonged immunomodulatory changes, albeit not exclusive to RFA treatment.

Pancreatic cancer and extrahepatic cholangiocarcinoma are malignancies with poor prognosis. As most patients with these conditions are diagnosed at an advanced stage, surgical resection is rarely possible. Additionally, such malignancies are typically resistant to current chemotherapy and radiation protocols [1]. With the search for improved outcomes and the development of endoluminal RFA catheters, endoscopic RFA has emerged as a promising palliative treatment option for pancreatic cancers and extrahepatic cholangiocarcinoma. Various retrospective series and novel randomised trials provide compelling evidence that such an intervention improves survival and prolongs stent patency [3,6,7,8,9,10].

RFA causes thermal damage created by a high-frequency alternating current released from an electrode into tissue. Temperatures greater than 50 °C lead to coagulative necrosis and cell death. Necrosis results in a loss of plasma membrane integrity. Early experimental studies suggested that RFA provides the immune system with an antigen source for antitumour immunity. Murine, rabbit, pig, and human models have demonstrated that thermal ablation generates a marked local inflammatory response, a dense T-cell infiltrate, and even a systemic tumour-antigen-specific T-cell response [12,13,21,24]. The hypothesis that localised treatment can have a wider systemic impact is known as the “abscopal effect”, a term first used in relation to local radiation [25]. The abscopal effect has also been implicated in the regression of metastasis following RFA of renal cancer [26].

In our study, we investigated the effects of endoluminal RFA in patients with pancreatic cancer and cholangiocarcinoma. We applied an extensive screen comprising a panel of 18 soluble markers and performed detailed immunophenotyping of peripheral lymphocytes and monocytes over a 30 day period post-ablation. First, we investigated the effect of RFA on the immune system of treated patients only, observing several changes. On the cytokine level, we found an early increase in IL-6 and a delayed increase in CXCL11, CXCL5, and CXCL1. IL-6, which is a pro-inflammatory cytokine produced by macrophages and parenchymal cells in response to damage and stimulates production of acute-phase proteins and neutrophils in the bone marrow. An increase in IL-6 in the hours and days following thermal ablation has been repeatedly documented [27,28,29]. IL-6 plays an important role in T-cell proliferation, survival, and trafficking to the tumour site [30]. Interestingly, it has also been suggested that IL-6 plays a negative role in suppressing the antitumour immune response by hindering the development and maturation of dendritic cells, increasing myeloid-derived suppressor cells (MDSCs), and activating the JAK/STAT3 signalling pathway [30]. While CXCL11 is a chemokine that mainly attracts activated T cells, allowing them to migrate to tumour sites and suppress tumours, CXCL5 and CXCL1 are chemokines with chemotactic and neutrophil-activating functions. Notably, both cytokines have also been implicated in tumour progression by promoting growth, invasion, and metastasis [31,32].

On the cellular level, we observed a delayed and prolonged stimulation of lymphocytes, perhaps attributable to an increase in total T cells (CD3+ cells) and an accompanying decrease in relative B cells (CD19+). The number of CD8+ cells was not affected early after RFA but increased by the 2 week mark, with a similar but less pronounced rise also occurring in CD4+ cells. We also observed an increase in NK cells on day 14. These findings are similar to those reported by other studies. For example, Giardino et al., who followed patients after pancreatic RFA, attributed a significant increase in the numbers of both CD8+ and CD4+ cells between day 3 and 30 following RFA to the activation of adaptive immunity [14].

In order to confirm that the observed changes resulted from RFA exclusively, we monitored the same parameters in a control group of untreated patients who underwent ERCP with stenting only. This type of control group has been typically absent from previous studies. For example, the abovementioned study by Giardino et al., which followed 10 patients who underwent surgical RFA via laparotomy, employed no control group [14]. Tellingly, we found that many of the changes recorded in the RFA-treated group also occurred in the control group, an observation corroborated by other authors. A well-designed study by Hinz et al. compared 10 patients with liver metastasis treated with RFA to eight patients treated by surgery. They found a statistically significant and marked elevation of IL-6 in the RFA group. However, this effect was even more pronounced in the control group, with no significant differences observed between the differentially treated patients [33].

Our controlled analysis prioritised changes that could be attributed specifically to RFA. First, on day 14 in treated patients, there was a statistically significant increase in the levels of CXCL1, which returned to normal levels by day 30. CXCL1 (previously GRO1) is a cytokine belonging to the CXC chemokine family that binds and attracts various immune cells, especially neutrophils, to the site of damage. An acute increase in IL-6 is best interpreted as a natural response to tissue damage. In contrast, however, the delayed upregulation of chemokines is understood to be associated with either the mobilisation of immune responses against the tumour or with reparation processes associated with tissue injury. Additionally, CXCL1 has been linked to cancer migration and invasion [34] as well as tumour progression and poor survival [35] in various cancer types. CXCL1 activates the G protein-coupled receptor (GPCR) CXCR2 [36]. Many GPCR agonists have been shown to induce transactivation of the epidermal growth factor receptor (EGFR), whereby crosstalk between chemokines and growth factor pathways induces proliferation [37].

Some studies investigating the effects of RFA on the immune system highlight its potentially negative role. Increased expression of HSP70 has been reported in various cancer types post-RFA [38,39]. HSP70 is known to chaperone tumour antigens to dendritic cells which, in turn, present antigens to T cells [40]. In contrast, heat shock proteins are overexpressed in numerous human cancers and, accordingly, implicated in cancer proliferation, invasion, and metastasis. This negative effect has also been recently connected to RFA. Gao et al. demonstrated increased expression of HSP70 in residual cancer cells post-RFA treatment as well as HSP70 promotion of pancreatic cancer growth via the activation of AKT-mTOR signalling [41]. Surprisingly, several reports have shown that RFA causes rapid tumour progression [42,43], one possible explanation being incomplete ablation of the targeted tumour. In a recent study, Shi et al. found that incomplete RFA promotes tumour progression through induction of sustained local inflammation and a predominance of myeloid suppressor cells, where the CCL2 chemokine plays a critical role [44]. In another study, RFA accelerated residual tumour progression by increasing tumour-infiltrating myeloid-derived suppressor cells and reducing the T-cell-mediated antitumour immune response [45]. The likelihood is that incomplete ablation is particularly relevant to endoluminal RFA, given the tumour is ablated from the bile duct with limited penetration to a depth of a few millimetres irrespective of tumour width.

In our controlled model, the changes we observed on a cellular level implicate the role of RFA in inducing inflammation. The number of immunosuppressive CD206+ monocytes declined on the first day after RFA compared to the controls. Basal values were reached at two weeks, remaining unchanged until one-month post-RFA. This phenotypic pattern in changes to blood monocytes seems to be directly affected by RFA, potentially reflecting either the effect of pro-inflammatory cytokines released in response to RFA or the recruitment of monocyte subpopulations at the site of injury. The latter explanation is consistent with observations by Shi et al., who found a higher proportion of immunosuppressive M2-like (CD11b+F4/80+CD206+) macrophage subsets in residual tumours post-RFA ablation [44].

The delayed downregulation of CD47 in the RFA group compared to the controls may reflect the higher sensitivity of this monocyte subpopulation to apoptosis. However, the number of monocytes affected was rather low to be clinically relevant. Recently, Svachova et al. reported a decrease in CD47 expression by blood monocytes following kidney allograft transplantation [46].

There are several possible explanations for the immune system alterations in our control group. ERC with biliary stent placement is an invasive procedure that can be associated with several physiological and immunological changes. Notably, all of our patients had had biliary obstruction, which was subsequently relieved by ERC. Nonetheless, the presence of remaining subclinical obstruction, including cholangitis, cannot be fully ruled out. All of our patients displayed an advanced malignancy, and a hallmark of cancer is prolonged inflammation. Thus, cancer is typically associated with representative changes in the systemic immune system. For example, advanced stages of pancreatic cancer are typically characterised by a reduction in total lymphocytes [47] along with elevated neutrophil frequencies in the blood (measured by the neutrophil-to-lymphocyte ratio). Both of these changes are associated with poor prognosis in multiple cancers including pancreatic cancer [48]. Importantly, we did not examine the phenotypic behaviour of cell populations. Considering that immune cells, such as T and NK cells, exhibit altered phenotypes in cancer, the numerical changes we observed may not have reflected their functional status [49].

Our study may also underscore differences between pancreatic cancer and cholangiocarcinoma, two diseases that share numerous similarities such as genetic alterations, late diagnosis, resistance to chemoradiation and unfavourable prognosis. We found no significant differences in the immune response to RFA except for slight differences in the extent of some changes. Intriguingly, the histological type of tumour exposed to RFA is believed to influence the immune response. Most of the published literature has investigated the impact of RFA on hepatocellular cancer, a tumour considered highly immunogenic [50] and for which RFA is a well-established treatment modality [2]. However, pancreatic cancer is characterised by a highly immunosuppressive microenvironment with only a sparse neoantigen-presenting capacity. These conditions likely contribute, for example, to the ineffectiveness of checkpoint inhibitors in recruiting more T cells [51]. Therefore, presenting more antigens as a consequence of RFA may not be sufficient to trigger a strong systemic response.

Where RFA does have an impact on the host immune system, it is likely to occur locally at the tumour site. Interestingly, there is persuasive evidence that the systemic immune response may be weak or undemonstrative in systemic circulation. Yet, changes may still be induced and effective locally. For example, in one preclinical murine model, no significant changes were found in systemic circulation, but there was a significant increase in tumour-infiltrating CD8+ cells along with a decrease in regulatory T cells post-RFA treatment [41]. Furthermore, many animal studies, for which tumours and tissues are readily available, have omitted any analysis of changes in systemic circulation [44,45,52].

In clinical practice, local recurrence or progression often eventually occurs following RFA treatment. Thus, the possibilities of combination treatment have been explored. Some immunology studies report that the limited effects of RFA on the immune system may be enhanced by additional treatments. For example, combination treatment involving RFA and sunitinib has been shown to activate the local tumour-specific antigen immune response, increase the frequency of CD8+ T cells, memory CD8+ T cells, and dendritic cells, and decrease the frequency of regulatory T cells. Another study found that RFA caused PD-1 upregulation in tumour-infiltrated T cells by enhancing hepatocyte growth factor expression, which was suppressed by sunitinib treatment [19]. In another study, a combination of RFA and local immunostimulation with a thermosensitive hydrogel containing GM-CSF-BCG induced a strong specific immune response, resulting in complete resolution of microscopic secondary colorectal cancer liver lesions in mice. Yet, another report found that the addition of systemic anti-PD1 enhanced the systemic-specific immune response and led to a complete regression of distant tumours in the majority of mice [52].

While our findings are compelling, there are numerous limitations to our study. First, we investigated a selection of immune system parameters at certain time points, meaning some changes may have been missed. For example, we omitted regulatory T cells and dendritic cells from our analysis. Second, our study only involved a limited number of patients, albeit more than most studies performed to date. Nevertheless, since the majority of our analysis revealed negative results (for which our study was not a priori powered), there is the possibility that positive results were also overlooked. Third, we examined changes in systemic circulation only. Endoluminal RFA may not be adequate at inducing measurable changes in the periphery beyond the local antitumour response. Fourth, endoluminal biliary RFA can potentially result in incomplete ablation, especially in the case of pancreatic cancer. Thus, our outcomes may have been influenced by incomplete ablation, at least in some patients. Fifth, our RFA system lacks temperature control. Temperature-controlled ablation enables to avoid high temperatures, resulting in increased impedance [53]. Despite these shortcomings, our study is the first to explore the immune response in a well-controlled human model of pancreatic cancer and cholangiocarcinoma.

In summary, our study provides insights into the impact of endoluminal RFA delivered through an endoscopically placed catheter on the systemic immune system in patients with pancreatic cancer and cholangiocarcinoma. Although we documented numerous changes in the levels of cytokines and chemokines as well as long-term changes in lymphocyte subpopulations, we were unable to attribute them specifically to RFA. Rather, it seems likely that these changes reflect the given endoscopic procedure along with the character and course of the type of malignant disease with biliary obstruction. The increase in CXCL1 and decrease in CD206+ cells would seem to reflect acute inflammation; however, this may also contribute to cancer progression. Further studies involving a larger number of patients are required in order to define the precise role of the immune system after endoluminal RFA including clinical studies aimed at evaluating any possible effects on survival.

## Figures and Tables

**Figure 1 biomedicines-10-01331-f001:**
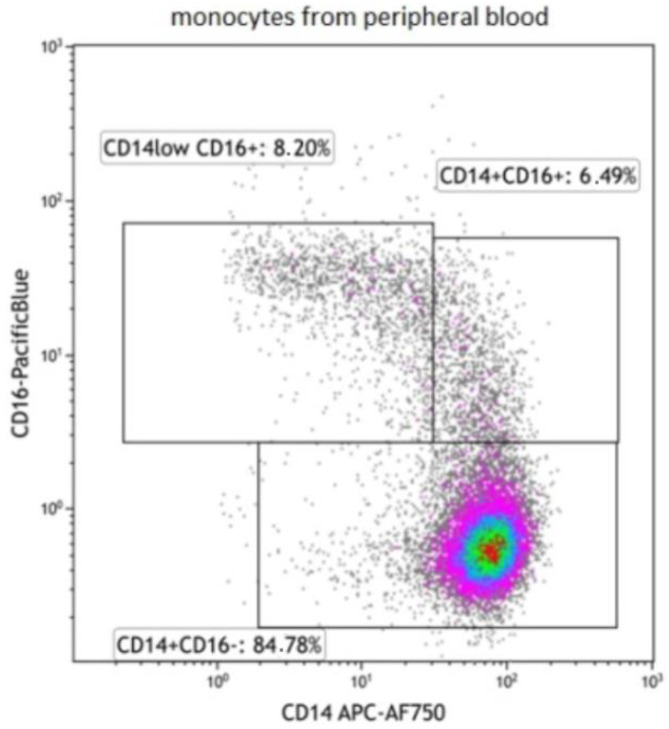
Gating strategy of monocyte subpopulations.

**Figure 2 biomedicines-10-01331-f002:**
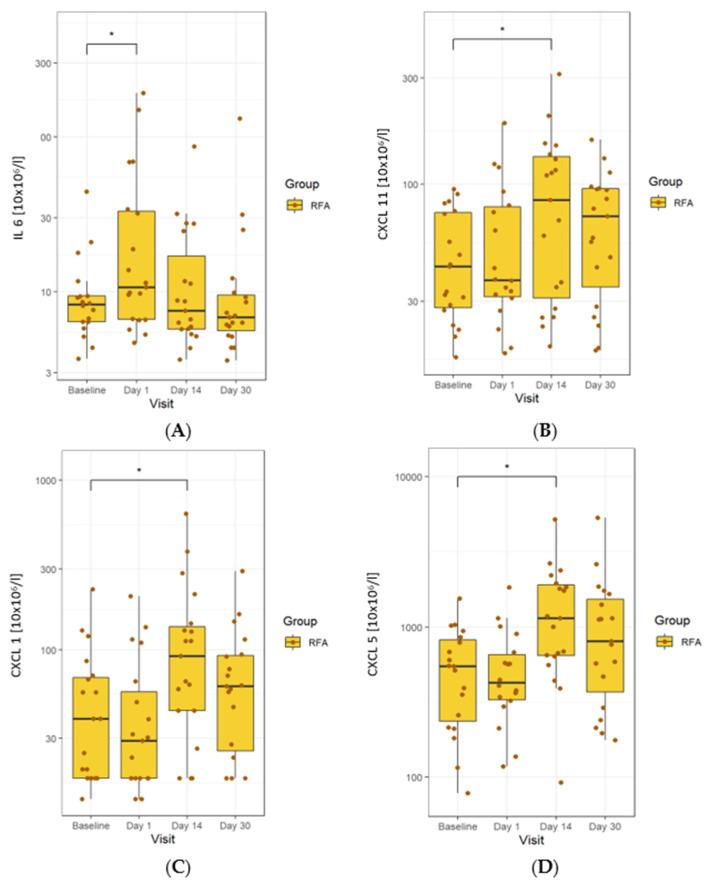
Changes in serum cytokines observed in the RFA-treated group. (**A**) levels of IL-6 were elevated early in the course of treatment and later returned to normal. (**B**) Levels of CXCL11 were elevated on day 14, which continued until the end of the study. (**C**) CXCL1 increased on day 14 and was then reduced on day 30. (**D**) CXCL5 rose on day 14 and remained elevated until day 30. The line in the middle of the box represents the median; the upper and lower ends of the box represent the first and third quartiles, respectively, excluding outlines. The upper and lower whiskers show the maximum and minimum scores, respectively. Individual dots represent individual patients’ values. Asterisks show the statistical significance.

**Figure 3 biomedicines-10-01331-f003:**
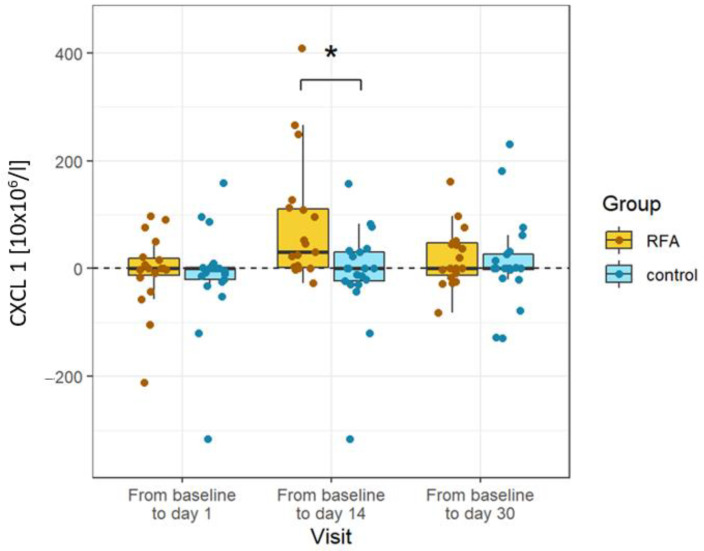
Changes in cytokines specific to RFA. CXCL1 was significantly elevated in the RFA-treated patients compared to the controls on day 14. Absolute differences from baseline are plotted for different visits using box and whisker plots. The line in the middle of the box represents the median; the upper and lower ends of the box represent the first and third quartiles, respectively. The upper and lower whiskers show the maximum and minimum scores, respectively, excluding outliers. Individual dots represent individual patients’ values. Asterisks show the statistical significance.

**Figure 4 biomedicines-10-01331-f004:**
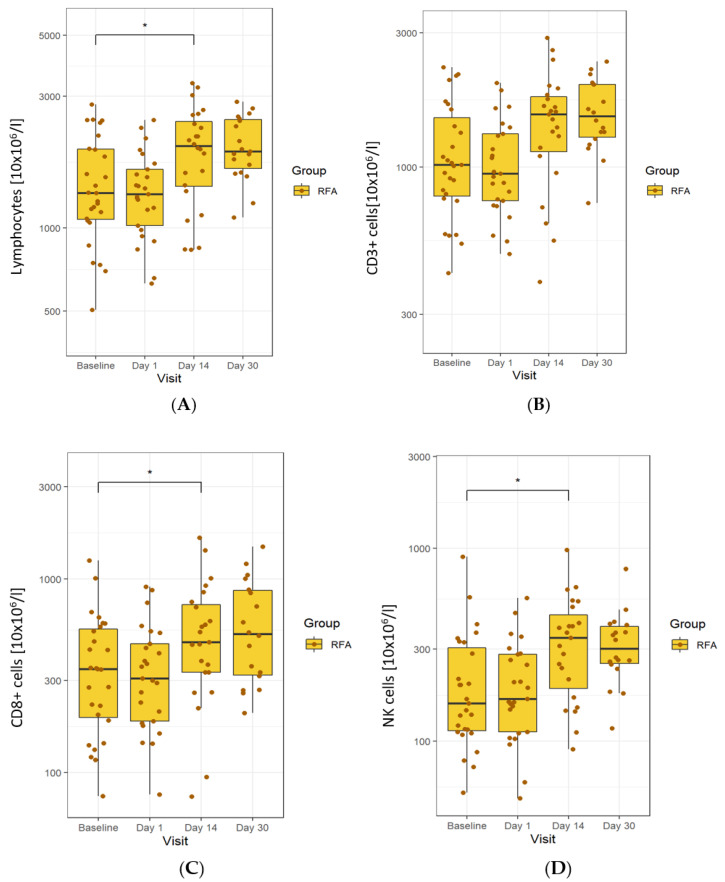
Changes in cells observed in the RFA-treated group. (**A**) An increase in the total lymphocytes on day 14 until day 30. (**B**) A rise in total T cells. (**C**) An increase in the number of CD8+ cells on day 14 until day 30. (**D**) An increase in NK cells on day 14, remaining until the end of study. The line in the middle of the box represents the median; the upper and lower ends of the box represent the first and third quartiles, respectively, excluding outliers. The upper and lower whiskers show the maximum and minimum scores, respectively. Individual dots represent individual patients’ values. Asterisks show the statistical significance. The *y*-axis is in logarithmic scale.

**Figure 5 biomedicines-10-01331-f005:**
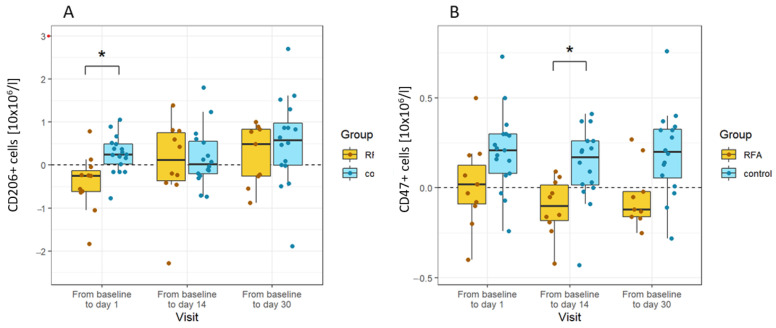
Changes in cells specific to RFA in CCC patients. (**A**) The number of CD206+ cells decreased shortly after the procedure in the RFA-treated patients compared to the controls (day 1). (**B**) The number of CD47+ was greater than the controls at all time points, and the difference was statistically significant on day 14. Absolute differences from baseline are plotted for different visits using box and whisker plots. The line in the middle of the box represents the median; the upper and lower ends of the box represent the first and third quartiles, respectively, excluding outliers. The upper and lower whiskers show the maximum and minimum scores, respectively. Individual dots represent individual patients’ values. Asterisks show the statistical significance. The *x*-axis shows the fold difference between values at each visit vs. baseline values before treatment.

**Table 1 biomedicines-10-01331-t001:** Summary of monoclonal antibodies used for measurement.

Antigen	Fluorochrome	Clone	Producer
tetraCHROME CD45-FITC/CD4-PE/CD8-ECD/CD3-PC5			Beckman Coulter, Brea, CA, USA
tetraCHROME CD45-FITC/CD56-PE/CD19-ECD/CD3-PC5			Beckman Coulter, Brea, CA, USA
CD16	PE	3G8	Beckman Coulter, Brea, CA, USA
HLA-DR	PE	Immu-357	Beckman Coulter, Brea, CA, USA
CD14	APC-AF 750	RM052	Beckman Coulter, Brea, CA, USA
CD16	Pacific Blue	3G8	Beckman Coulter, Brea, CA, USA
HLA-DR	PC7	Immu-357	Beckman Coulter, Brea, CA, USA
CD163	PE	GHI/61	BioLegend, San Diego, CA, USA
CD206	APC	15–2	BioLegend, San Diego, CA, USA
CD209	PerCP/Cy5.5	9E9A8	BioLegend, San Diego, CA, USA
CD47	FITC	MEM-122	Beckman Coulter, Brea, CA, USA

**Table 2 biomedicines-10-01331-t002:** Summary of detected cytokines and chemokines by Luminex technology.

	Detected Markers
Proinflammatory cytokines	IL-1α, IL-1β, IL-6, IL-18, IL-33, IL-36β, TNFα
Anti-inflammatory cytokines	IL-1RA, IL-10
Chemokines attracting mainly neutrophils	CXCL1/GROα, CXCL5/ENA-78, CXCL8/IL-8
Chemokines attracting mainly T lymphocytes	CXCL9/MIG, CXCL10/IP-10, CXCL11/I-TAC, CCL21/6Ckine
Chemokines attracting mainly monocytes, NK cells, eosinophiles, and basophiles	CCL2/MCP-1, CCL5/RANTES

**Table 3 biomedicines-10-01331-t003:** Characteristics of the study population.

	PC		CCC	
	RFA	Controls	RFA	Controls
N	11	8	8	13
Mean age (SD)	73.6 (9.9)	69.0 (6.2)	63.3 (7.3)	67.1 (9.0)
Males/females	7/4	4/4	5/3	9/4
Bismuth classification	NA	NA		
Type I			1	1
Type II			1	4
Type III			4	4
Type IV			1	3
TNM stage				
Stage I	3	0	0	1
Stage II	4	2	1	1
Stage III	2	2	3	7
Stage IV	2	4	4	4
Prior chemotherapy	1	2	2	3

## Data Availability

Data is contained within the article and Appendix A.

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
