# Peer review of "Impact of Endoluminal Radiofrequency Ablation on Immunity in Pancreatic Cancer and Cholangiocarcinoma"

_biomedicines, 2022, doi:10.3390/biomedicines10061331_

Round 1

Reviewer 1 Report

Line 42 correct “is” for “are”

Paragraph between lines 41 and 43 needs a reference.

Line 54 change the word “instigates” for a more suitable word.

Paragraph between lines 53 and 57 requires a reference.

Line 62 Replace the word “implicated”.

Line 93 says “proportion of consecutive patients” It should be convenient to include the percentage of cases.

Line 156 Revise syntax: eliminate “then”

Figures 2 and 4 and 5 are too small and impossible to analyze.

Lines 286-287 “As most patients with these conditions are diagnosed at an advanced stage,  surgical resection is rarely justified” I would change the sense of this phrase for

“As most patients with these conditions are diagnosed at an advanced stage, surgical resection is rarely possible.”

Lines 287-288 “Additionally, such malignancies are typically re- 287 sistant to current chemotherapy and radiation protocols”. This needs a reference.

Line 296. Change the word “instigates”

Line 322 Reference is missing.

Lines 325-326 Revise syntax

Line 332 Revise syntax

“Washing procedure” is more appropriate than “wash procedure”.

Author Response

To the editor of Biomedicines

We would like to thank you and your reviewers for very valuable review of our manuscript.  We have made all the requested changes suggested by you reviewers.  Detailed point by point response can be found below.  We hope that our manuscript will now be suitable for the audience of your journal.

Yours sincerely,

Jana Jarosova and Tomas Hucl

Point by point response to reviewers

Reviewer 1

Line 42 correct “is” for “are”

Requested change has been done.

Paragraph between lines 41 and 43 needs a reference.

Reference has been included.

Line 54 change the word “instigates” for a more suitable word.

Requested change has been done.

Paragraph between lines 53 and 57 requires a reference.

Reference has been included.

Line 62 Replace the word “implicated”.

The word has been replaced.

Line 93 says “proportion of consecutive patients” It should be convenient to include the percentage of cases.

Requested change has been done.  The percentage has been included.

Line 156 Revise syntax: eliminate “then”

Requested change has been done

Figures 2 and 4 and 5 are too small and impossible to analyze.

The figures have been enlarged.

Lines 286-287 “As most patients with these conditions are diagnosed at an advanced stage,  surgical resection is rarely justified” I would change the sense of this phrase for

“As most patients with these conditions are diagnosed at an advanced stage, surgical resection is rarely possible.”

Lines 287-288 “Additionally, such malignancies are typically re- 287 sistant to current chemotherapy and radiation protocols”. This needs a reference.

Reference has been included.

Line 296. Change the word “instigates”

Requested change has been done

Line 322 Reference is missing.

Reference has been included.

Lines 325-326 Revise syntax

Syntax has been revised.

Line 332 Revise syntax

Syntax has been revised.

“Washing procedure” is more appropriate than “wash procedure”.

Requested change has been done.

Reviewer 2 Report

The author observed an early increase in IL-6 levels and a delayed increase in CXCL1, CXCL 5 and CXCL 11 levels as well as an increase in CD8+ and NK cells. Explicitly in response to RFA, we observed a delayed increase in serum CXCL1 levels and an early decrease in the number of anti-inflammatory CD206+ blood monocytes. Albeit, the author provides an insight into the exploration of potential therapeutic targets for cancer, I still have some suggestions.

1, All figures are highly professional, and the authors should guide the readers to the meaning of the images appropriately; otherwise, it is likely to cause misunderstandings. Therefore, I suggest that the author consider revising these figure legends again.

2, Since RFA is increasingly used as a palliative treatment for pancreatic cancer and cholangiocarcinoma. Therefore, it is worthy to further confirm or validate Table 2 via public database, including prognoscan or proteinatlas (PMID: 19393097, 25613900, 32064155, 34172056, 35326643).

3, So far, the tumor infiltrates immune cells is vital for patient survival. It is worthy of exploring the above genes that correlate with immune cells by using the "TIMER" (http://timer.cistrome.org/) analysis tool (PMID: 32442275, 34329194, 34794429, 35330401, 35203508).

Author Response

To the editor of Biomedicines

We would like to thank you and your reviewers for very valuable review of our manuscript.  We have made all the requested changes suggested by you reviewers.  Detailed point by point response can be found below.  We hope that our manuscript will now be suitable for the audience of your journal.

Yours sincerely,

Jana Jarosova and Tomas Hucl

Point by point response to reviewers

Reviewer 2

The author observed an early increase in IL-6 levels and a delayed increase in CXCL1, CXCL 5 and CXCL 11 levels as well as an increase in CD8+ and NK cells. Explicitly in response to RFA, we observed a delayed increase in serum CXCL1 levels and an early decrease in the number of anti-inflammatory CD206+ blood monocytes. Albeit, the author provides an insight into the exploration of potential therapeutic targets for cancer, I still have some suggestions.

1, All figures are highly professional, and the authors should guide the readers to the meaning of the images appropriately; otherwise, it is likely to cause misunderstandings. Therefore, I suggest that the author consider revising these figure legends again.

The figure legends have been revised as suggested.

2, Since RFA is increasingly used as a palliative treatment for pancreatic cancer and cholangiocarcinoma. Therefore, it is worthy to further confirm or validate Table 2 via public database, including prognoscan or proteinatlas (PMID: 19393097, 25613900, 32064155, 34172056, 35326643).

We thank the reviewer for this valuable comments and we have included now analyses of public database for expression and prognosis correlation.  We had to use a different online tool than those suggested because the suggested tools do not contain datasets of pancreatic cancer and/or cholangiocarcinoma.

3, So far, the tumor infiltrates immune cells is vital for patient survival. It is worthy of exploring the above genes that correlate with immune cells by using the "TIMER" (http://timer.cistrome.org/) analysis tool (PMID: 32442275, 34329194, 34794429, 35330401, 35203508).

Again, we thank the reviewer for his suggestion.  We have now added an analysis of immune cell infiltration performed using the TIMER tool.
